# The MiR-320 Family Is Strongly Downregulated in Patients with COVID-19 Induced Severe Respiratory Failure

**DOI:** 10.3390/ijms221910351

**Published:** 2021-09-26

**Authors:** Ruth P. Duecker, Elisabeth H. Adam, Sarah Wirtz, Lucia Gronau, Yascha Khodamoradi, Fabian J. Eberhardt, Helena Donath, Desiree Gutmann, Maria J. G. T. Vehreschild, Kai Zacharowski, Hermann Kreyenberg, Andreas G. Chiocchetti, Stefan Zielen, Ralf Schubert

**Affiliations:** 1Division for Allergy, Pneumology and Cystic Fibrosis, Department for Children and Adolescence, University Hospital Frankfurt, Goethe-University, 60590 Frankfurt am Main, Germany; Sarah.Wirtz@kgu.de (S.W.); Lucy.Gronau@kgu.de (L.G.); Helena.Donath@kgu.de (H.D.); Desiree.Gutmann@kgu.de (D.G.); stefan.zielen@kgu.de (S.Z.); Ralf.Schubert@kgu.de (R.S.); 2Department of Anaesthesiology, Intensive Care Medicine and Pain Therapy 2, University Hospital Frankfurt, Goethe-University, 60590 Frankfurt am Main, Germany; Elisabeth.Adam@kgu.de (E.H.A.); Kai.Zacharowski@kgu.de (K.Z.); 3Department of Food Technology, University of Applied Sciences Fulda, 36037 Fulda, Germany; 4Department of Internal Medicine, Infectious Diseases, University Hospital Frankfurt, Goethe-University, 60590 Frankfurt am Main, Germany; Yascha.Khodamoradi@kgu.de (Y.K.); eberhardt.fabian@googlemail.com (F.J.E.); Maria.Vehreschild@kgu.de (M.J.G.T.V.); 5Division for Stem Cell Transplantation and Immunology, Department for Children and Adolescence, University Hospital Frankfurt, Goethe-University, 60590 Frankfurt am Main, Germany; Hermann.Kreyenberg@kgu.de; 6Department of Child and Adolescent Psychiatry, Psychosomatics and Psychotherapy, University Hospital Frankfurt, Goethe-University, 60590 Frankfurt am Main, Germany; Andreas.Geburtig-Chiocchetti@kgu.de

**Keywords:** miRNA, SARS-CoV-2, lung disease, respiratory failure, D-dimer, CRP

## Abstract

A high incidence of thromboembolic events associated with high mortality has been reported in severe acute respiratory syndrome coronavirus type 2 (SARS-CoV-2) infections with respiratory failure. The present study characterized post-transcriptional gene regulation by global microRNA (miRNA) expression in relation to activated coagulation and inflammation in 21 critically ill SARS-CoV-2 patients. The cohort consisted of patients with moderate respiratory failure (*n* = 11) and severe respiratory failure (*n* = 10) at an acute stage (day 0–3) and in the later course of the disease (>7 days). All patients needed supplemental oxygen and severe patients were defined by the requirement of positive pressure ventilation (intubation). Levels of D-dimers, activated partial thromboplastin time (aPTT), C-reactive protein (CRP), and interleukin (IL)-6 were significantly higher in patients with severe compared with moderate respiratory failure. Concurrently, next generation sequencing (NGS) analysis demonstrated increased dysregulation of miRNA expression with progression of disease severity connected to extreme downregulation of miR-320a, miR-320b and miR-320c. Kyoto encyclopedia of genes and genomes (KEGG) pathway analysis revealed involvement in the Hippo signaling pathway, the transforming growth factor (TGF)-β signaling pathway and in the regulation of adherens junctions. The expression of all miR-320 family members was significantly correlated with CRP, IL-6, and D-dimer levels. In conclusion, our analysis underlines the importance of thromboembolic processes in patients with respiratory failure and emphasizes miRNA-320s as potential biomarkers for severe progressive SARS-CoV-2 infection.

## 1. Introduction

The ongoing severe acute respiratory syndrome coronavirus 2 (SARS-CoV-2) global pandemic is one of the greatest threats to our health and to economic and social stability worldwide. The course of the disease shows a wide diversity, ranging from often completely symptom-free to mild and moderate progression with pneumonia or even lung failure occurring in around 10% of cases [1,2]. Advanced age and comorbidities, such as obesity, diabetes mellitus and cardiovascular diseases, have been identified as risk factors for a severe course and death [3,4,5].

The severe course of SARS-CoV-2 infection with lung failure is usually triggered by sepsis, pneumonia, or organ failure [1,6]. According to Chinese studies, the occurrence of acute respiratory distress syndrome (ARDS) in the context of COVID-19 is of decisive importance for the prognosis [3,7]. Histopathological, diffuse alveolar damage on the basis of direct, viral cytotoxic effects on pneumocytes, and an accompanying inflammatory reaction are described [4,5]. Many patients with severe ARDS show massively increased levels of leukocytes, C-reactive protein (CRP), interleukin (IL)-6, and D-dimers in response to the viral infection [5]. A cytokine storm, found mainly in severely ill patients, can contribute to thrombogenicity and multiple organ failure with fatal consequences. Many publications report an unusually high incidence of thromboembolic events, including pulmonary embolism and parenchymal abnormalities, which significantly increase mortality [8,9,10]. In a recent case series of autopsies, thrombotic events were observed in at least one major organ in all autopsies, predominantly in the lung (89%), heart (56%), and kidneys (44%) of nine patients [11].

Numerous studies in immune-mediated lung diseases such as asthma, chronic inflammatory lung disease (COPD) or ARDS indicate that epigenetic regulation contributes to the balance of the immune response to external and internal stimuli [12,13,14]. This in turn suggests that an aberrant epigenetic modification could lead to immune disorders and thus to the development of inflammatory and fibrotic lung diseases. A central epigenetic mechanism is the post-transcriptional regulation by so-called microRNAs (miRNAs) [15]. MiRNAs are small, non-coding RNAs that bind to target messenger RNAs (mRNAs) in order to control gene expression post-transcriptionally. Thus, miRNAs play an important role in the control of a large number of cellular processes (e.g., cell proliferation, cell differentiation, and apoptosis) [16]. In addition to their role in chronic lung disease, miRNAs also modulate an acute phase of inflammation, making them a central key to understanding SARS-CoV-2 infection [17].

## 2. Results

Twenty-one patients (median age 72 years, range 48–91 years) with SARS-CoV-2 infections were divided in two groups on the basis of their supply with oxygen and ventilation (Table 1). The first group consisted of patients with moderate respiratory failure (*n* = 11) with a median age of 76 years (range 48–91 years), nine males and two females. The second group consisted of ten patients with severe respiratory failure (*n* = 10), with a median age of 69 years (range 52–79 years), including eight males and two females. All patients with severe respiratory failure were intubated and nine out of ten received non-invasive ventilation, whereas only three patients out of eleven in the moderate respiratory failure group received non-invasive ventilation and none of them were intubated. Four patients from the group with severe respiratory failure died during the treatment. No significant differences in comorbidities were seen between patients with moderate and severe respiratory failure (Table 1). The miRNA expression profiles of these groups were compared to those of an age-matched healthy control group (median age 68, range 49–89 years, five males and three females). The control group had no comorbidities and did not receive any treatment. 

### 2.1. Markers of Inflammation and Coagulation Are Increased in COVID-19 Patients with Severe Respiratory Failure

To assess the level of inflammation, CRP, IL-6 and D-dimers and the activated partial thromboplastin time (aPTT) were measured (Figure 1). CRP and IL-6 concentrations were significantly higher in the blood of patients with severe respiratory failure compared to patients with moderate respiratory failure, while in this group the highest values were found at acute onset.

D-dimers and aPTT values showed a steady increase from the acute stage of the disease to 7 days later in patients with moderate respiratory failure from the acute stages of the disease to 7 days later in patients with severe respiratory failure. This is especially evident for the levels of D-dimers, which were found to be significantly higher in the course of disease in patients with severe respiratory failure compared to all other visits.

### 2.2. MiR-320 Family Is Strongly Downregulated in COVID-19 Patients with Severe Respiratory Failure

MiRNA expression in whole blood of COVID-19 patients with moderate and severe respiratory failure at acute stage of disease and 7 days later were compared with the miRNA profiles of age-matched healthy individuals (Figure 2a). In total, 853 miRNAs were detected by NGS. Comparison revealed an increase in the number of significantly dysregulated miRNAs (*p*.adj < 0.05) depending on visit (acute or seven days later) and group (moderate vs severe) (Figure 2b). Compared to the matched controls, patients with moderate respiratory failure showed 27 (3.2%) dysregulated miRNAs at acute stage of disease and 129 (15.1%) dysregulated miRNAs 7 days later. Patients with severe respiratory failure exhibited 345 (41.7%) dysregulated miRNAs at the acute stage of the disease and 419 (49.1%) dysregulated miRNAs 7 days later. Further, these patients showed a significantly different expression pattern than controls, with 227 upregulated and 192 downregulated miRNAs (Figure 2c). Of particular note is the extreme significance in the downregulation of the miR-320 family members including miR-320a (*p*.adj = 1.51 × 10^−32^), miR-320b (*p*.adj = 4.19 × 10^−21^) and miR-320c (*p*.adj = 5.45 × 10^−14^) (Figure 2d). The expression was reduced to −2.67log2 (fold-change) for miR-320a, −2.65log2 (fold-change) for miR-320b, and −2.35log2 (fold-change) for miR-320c in patients with severe respiratory failure compared to controls (Figure 2e). The most strongly upregulated miRNAs were miR-374a-3p (log2FC 5.84), miR-15a-3p (log2FC 5.42), miR-3688-5p (log2FC 4.97) and miR-4721 (log2FC 4.65). The most strongly downregulated miRNAs were miR-4747-3p (log2FC − 6.05), miR-4429 (log2FC − 6.03), miR-6729-3p (log2FC − 5.98) and miR-1908-5p (log2FC − 5.73). All significantly deregulated miRNAs of these patients are shown in Appendix A.

### 2.3. MiR-320 Family Is Involved in Inflammation and Endothelial Dysfunction

To identify the contribution of the miRNAs to biological processes, the KEGG-pathway enrichment analysis was performed, and targets of the individual miRNAs were assessed. Targets of miR-320a, miR-320b, and miR-320c were enriched in 13 pathways including inflammation, endothelial dysfunction and cancer (Figure 3a, Appendix A). With regard to inflammation and endothelial dysfunction, the analysis shows that the miR-320 family members are involved in the regulation of 20 genes within the Hippo signaling pathway (*p* = 8.41 × 10^−12^), 13 genes within the TGF-β signaling pathway (*p* = 5.16 × 10^−8^), and 12 genes as part of the regulation of adherens junctions and epithelial paracellular permeability (*p* = 5.62 × 10^−7^) (Figure 3b, Appendix A).

### 2.4. Protein–Protein Interaction in Inflammation and Endothelial Dysfunction

On the basis of the KEGG analysis, we used the STRING web server to visualize the network of the protein–protein interaction of the miRNA targets involved in the Hippo signaling pathway, TGF-β signaling pathway and regulation of adherens junctions (Figure 3c). The network analysis shows a strong interaction between the target proteins between the pathways (PPI enrichment *p* value: < 1.0 × 10^−16^), revealing three connecting targets, SMAD2, SMAD3 and TGFBR2. Further, four genes, SMAD7, PPP2CA, PPP2CB and MYC, are located in the Hippo signaling and in the TGF-β signaling pathway, two genes, CDH1 and CTNNB1, are involved in the Hippo signaling pathway and regulation of adherens junctions, and MAPK1 connects the TGF-β signaling pathway and the regulation of adherens junctions.

### 2.5. MiR-320 Family Members’ Expression Correlates with Inflammation and Coagulation

We next performed correlation analyses to investigate the relationship between miRNA expression and markers of inflammation and coagulation (Figure 4). Expression of all miR-320 family members significantly correlated with CRP concentration (miR-320a *p* < 0.0001, *r* = −0.5651, miR-320b *p* < 0.0001, *r* = −0.5839, miR-320c *p* = 0.0004, *r* = v0.4784, Figure 4a), with D-dimer levels (miR-320a *p* < 0.0001, *r* = −0.6682, miR-320b *p* < 0.0001, *r* = −0.6319, and miR-320c *p* < 0.0001, *r* = −0.6266, Figure 4b) and with the IL-6 concentration (miR-320a *p* < 0.0008, *r* = −0.4966, miR-320b *p* < 0.0015, *r* = −0.4753, miR-320c *p* = 0.002, *r* = −0.4626, Figure 4c) in the blood of COVID-19 patients and healthy controls. Although the correlation between the miR-320 family members and the number of neutrophils just missed significance, neutrophilic miRNAs, miR-142-3p (*p* < 0.0001, *r* = 0.5679) and miR-223-3p (*p* < 0.0001, *r* = 0.6496), significantly correlated with neutrophil counts in our patient cohorts (Appendix A).

## 3. Discussion

MiRNAs have emerged as critical regulators in the pathogenesis of inflammatory lung diseases, but they also play a crucial role in the immune response to respiratory viral infections [18,19,20]. Given the potential of miRNAs as biomarkers that affect post-transcriptional gene regulation, the present study aimed to characterize global miRNA expression profiles in the peripheral blood of patients with moderate to severe SARS-CoV-2 infection with respiratory failure.

The clinical and laboratory characterization of our cohorts of critically ill patients was in line with the finding of previous studies that severe SARS-CoV-2 infections may lead to excessive and uncontrolled inflammatory responses, which is associated with a massive release of inflammatory cytokines accompanied by vasculitic processes and endothelial dysfunction [21,22]. As described earlier, CRP and IL-6 concentrations were significantly higher in the blood of patients with severe disease when compared with patients with moderate disease progression [23]. In addition, D-dimer and aPTT values in our groups of patients supported the fact that coagulation dysfunction is more likely to occur in critically ill patients as observed in other infectious diseases [24].

Concurrent with these observations, the NGS analysis demonstrated an elevated dysregulation of miRNA expression with progression of disease severity. Thus, there was a significant increase in the number of dysregulated miRNAs in patients with COVID-19-induced severe respiratory failure. Among these up- and downregulated miRNAs, the members of the miR-320 family, miR-320a, miR-320b and miR-320c, showed the highest significance in downregulation and were negatively correlated with CRP, D-dimer and IL-6 concentrations. Down-regulation of the miRNAs-320 is already observed in patients with acute moderate symptoms and decreases steadily in patients with a severe course of the disease. The miRNA-320 family has already been described in patients with deep vein thrombosis (DVT) [25,26]. DVT is also a severe complication of COVID-19 patients and, therefore, the identification of high-risk COVID-19 patients is important to promptly initiate thrombosis prophylaxis [27]. In contrast to our findings, Jiang et al. reported a significant up-regulation of miRNA-320a and miRNA-320b correlating with D-dimer concentrations in DVT patients, but not in post-thrombotic syndrome [25]. These discrepancies could be explained by a protective mechanism against the sudden thrombosis in COVID-19 patients, which might lead to a drop in miRNA-320 expression. Indeed, as described by Zhu et al. downregulation of miR-320 exerts a protective effect on myocardial ischemia-reperfusion injury via promotion of Nrf2 expression [28]. Down-regulation of miR-320 has further been linked to the regulation of inflammation and oxidative stress by Nrf2 and miR-320 over-expression restrained TGF-β1 signaling, hindering inflammation and reactive oxygen species (ROS) production in a cellular model of diabetic retinopathy [29]. In line with this thinking, miR-320-5p was downregulated in acute pancreatitis (AP) serum and caerulein-treated AR42J cells, an in vitro cell model of AP [30]. Interestingly, overexpression of miR-320-5p resulted in a marked reduction in TNF-α, IL-1β, IL-6 and IL-8 levels by targeting tumor necrosis factor receptor-associated factor 3 (TRAF). The role of miR320 in the regulation of IL-6 and TNF-α has also been described in the inflammatory response in 3T3-L1 adipocytes and in smoke inhalation injury rats [31,32]. In addition, miR-320 expression is significantly decreased in the inflamed mucosa of patients with ulcerative colitis and Crohn’s disease [33]. An alternative explanation to this assertion would be the direct involvement of the virus in down-regulating miRs-320 expression. In this regard, a fluctuating expression of miRNAs including miR-320-a/b/c in adenovirus infected human lung fibroblast has been demonstrated by Zhao et al. [34]. This finding is in line with Ishida et al. who detected a down-regulation of miR-320 in Huh7 cells infected by hepatitis C virus [35].

Beside miR-320, there are many other miRNAs that were strongly deregulated in the patient populations. Of the most upregulated miRNAs, miR-374a-3p and miR-15a-3p are involved in acute kidney and neuroinflammation, respectively, whereas mir-4721 was proposed as one of the miRNA molecular diagnostic markers for liver fibrosis in patients with chronic hepatitis B [36,37,38]. Of the most downregulated miRNAs, miR-4747-3p was found upregulated in Alzheimer’s Disease and hsa-miR-4429 was suggested as a potential diagnostic biomarker for biliary atresia [39,40]. MiR-1908-5p plays an important role in the regulation of cardiometabolic phenotypes [41]. However, although there is very little or no literature about these miRNAs, they should be kept in mind as possible markers for progressive COVID-19 disease. Other miRNAs that were significantly deregulated in our analysis, such as miR-21-5p, miR-27, 126-5p, miR-146, or miR-142 have already attracted attention in other COVID-19 studies [42,43,44]. In agreement with Garg et al., who investigated circulating microRNAs in critically ill COVID-19 patients, we discovered dysregulated miRNAs involved in inflammatory processes such as miR-21-5p and 126-5p [44]. In line with de Gonzalo-Calvo et al., who investigated the circulating miRNA profile in hospitalized patients admitted to clinical wards without requiring critical care and patients admitted to the intensive care unit (ICU), we found an upregulation of miR-27a-3p, miR-27b-3p, and miR-148a-5p that distinguished between ICU and ward patients [43]. Furthermore, like Donyavi et al. we looked at cellular miRNAs in the acute and post-acute phase of COVID-19, we found miR-29a-3p and miR-146a-3p upregulated in our severe group of patients [42]. However, we could not confirm all the results of other studies. In contrast to the study by Tang et al., who detected decreased expression of miR-146a-5p, miR-21-5p, and miR-142-3p comparing moderate and severe COVID-19 patients with healthy controls, we saw an increase in these inflammatory miRNAs following disease severity [45]. With regard to neutrophilia, down-regulation of these inflammatory markers is surprising, in particular, down-regulation of miR-142-3p as a marker for neutrophilic granulocytes [46]. In fact, expression of both neutrophilic miRNAs, miR-142-3p and miR-223-3p, significantly correlated with neutrophil counts in our patient cohort. Overall, the results of the various studies differ from each other, in some cases considerably, in terms of the materials investigated, the sample preparation and the sequencing method, but also in terms of the severity of the patients examined concerning this matter.

Taken together, the miR-320 family seems to be downregulated in inflammatory processes and their expression could be directly influenced by viral infections. Our KEGG analysis and our correlation analysis supported the involvement of the miR-320s in inflammatory pathways. MiRs-320a/b/c have been computed to target components of the TGF-β and Hippo signaling pathways but also suggested a role in the regulation of adherens junctions. The corresponding targets, such as TGF-βR1, SMAD-proteins, LATS1/2 and β-catenin point to further fibrotic processes that might be central to COVID-19-induced lung fibrosing lesions [47]. In particular, the dysregulation of the TGF-β signaling pathway triggers an extensive immune reaction, leading to immense pulmonary damage and TGF-β levels in infected individuals and may thus predict poor outcomes in patients with COVID-19 [48,49]. In addition to TGF-β signaling, our analysis showed that LATS1, LATS2 and WWRT1 (TAZ) are core components of the Hippo pathway as targets of the miRNA-320 family. Activation of the Hippo pathway is regulated by cell polarity, cell adhesion and cell junction proteins [50]. Emerging evidence shows that Hippo signaling controls tissue homeostasis but is also responsible for immune cell recruitment and activation [51]. Disassembly of adherens junctions and tight junctions in alveolar epithelial cells and endothelial cells due to infection by SARS-CoV-2 results in remodeling and deposition of fibrin clots in the alveolar capillaries, leading to disintegration and thickening of the blood-gas barrier, and finally, hypoxia [52]. There is growing evidence that the Hippo pathway is involved in cytoskeletal remodeling, TGF-β signaling and the regulation of adherens junctions [53]. Furthermore, recent literature suggests that endothelial and epithelial-mesenchymal transition (EMT) might be central to COVID-19 lung fibrosing lesions [54]. In this context, Sugano et al. could show that suppression of the TGFβ receptor type-I (TGFBR1) gene, at least partly through the induction of miR-320a/b/d, leads to inhibition of EMT [55]. However, it is important to keep in mind that the results of the study are correlative in nature and target prediction was performed by computational analysis. Thus, further studies and experiments should be performed to validate our findings.

In conclusion, our study demonstrated a strong downregulation of the miR-320 family members, miR-320a, miR-320b and miR-320c. This correlates with markers of inflammation and coagulation in patients with COVID-19-induced moderate to severe respiratory failure. This study underlines endothelial dysfunction as a central feature of COVID-19 and sheds light on the miR-320-mediated pathomechanism. In addition, our in-silico analysis underlines the importance of therapies to manage lung fibrosis and its underlying molecular mechanisms in the pathologic pneumological processes and indicates miRNA-320s as potential early biomarkers for severe progressive respiratory failure in SARS-CoV-2 infection.

## 4. Materials and Methods

### 4.1. Patients

Clinical data such as medical history, body mass index (BMI), inflammatory parameters, duration of ventilation, co-morbidities, mortality, and samples for miRNA analysis were collected from eleven patients with moderate and ten patients with severe respiratory failure secondary to SARS-CoV-2 infection during their stay at the infectious disease ward or intensive care unit of the University Hospital Frankfurt. Data were obtained from 21 critically ill patients, consistent with patients with moderate respiratory failure (*n* = 11) and severe respiratory failure (*n* = 10) at the acute stage (day 0–3) and in the later course of the disease (>7 days). In addition, age-matched healthy volunteers were recruited as a control group. Blood samples using the PAXgene Blood RNA System for miRNA were taken from all patients and controls at both visits and stored at −20 °C until further analysis.

The study was approved by the Ethics Committee of the Goethe-University Frankfurt (20-836). Prior to the start of the study, written consent was obtained from all patients or authorized representatives and controls. A SARS-CoV-2 infection was identified by reverse transcription quantitative real time PCR (RT-qPCR) of oral/nasal swabs along with a clinical review.

### 4.2. Biomarker for Inflammation and Coagulation

Inflammatory biomarkers such as CRP, and IL-6 and the biochemical biomarker D-dimer and activated partial thromboplastin time (aPTT) were routinely analyzed at the central laboratory University Hospital Frankfurt, Goethe-University.

### 4.3. MiRNA Sequencing and Analysis

Total RNA, including miRNA, was isolated using the PAXgene Blood miRNA Kit (Qiagen, Hilden, Germany) according to the manufacturer’s instructions. Concentration of RNA was assessed using Nanodrop Lite spectrometry (Thermo Scientific, Dreieich, Germany) and RNA integrity (RIN) was assessed using the Agilent RNA 6000 Nano Kit and the Agilent 2100 Bioanalyzer (Agilent Technologies, Santa Carla, CA, USA). MiRNA libraries were generated with the QIAseq miRNA Library Kit (Qiagen, Hilden, Germany). Thereafter, cDNA concentration was determined using the Qubit dsDNA Assay Kit, Qubit assay tubes and the Qubit 3.0 fluorometer (all from Thermo Fisher Scientific, Dreieich, Germany). DNA quality was validated with the Agilent High Sensitivity DNA Reagents and the Agilent 2100 Bioanalyzer (Agilent Technologies, Santa Carla, CA, USA) using High-Sensitivity DNA chips. Libraries were frozen at −20 °C until miRNA sequencing. Next generation sequencing (NGS) was performed with the MiSeq Reagent Kit v3, the PhiX Sequencing Control v3 and the MiSeq™ Desktop Sequencer (all Illumina Inc., San Diego, CA, USA). Data were pre-processed using the Qiagen Web Portal service, which converted coverage files into raw counts matrices (https://geneglobe.qiagen.com/de/analyze accessed on 06/17/2021). Differential expression analysis was performed in RStudio 1.2.1335 (https://cran.r-project.org/ accessed on 06/17/2021) as described earlier [56]. Fdr correction was applied for each miRNA passing DESeq-quality thresholds and miRNAs were considered to be differentially expressed with *p*.adj < 0.05. miRNAs with *p*.adj < 0.001 were used for further analysis.

### 4.4. Pathway Enrichment Analysis of Target Genes and Protein–Protein Interaction Network

Kyoto encyclopedia of genes and genomes (KEGG) pathway analysis based on miRNA signatures was performed with DIANA-mirPath v.3.0 (TarBase v.7.0) and miRNet (miRTarBase v8.0) to detect pathways and targets regulated by the individual miRNAs [57]. Targets of the candidate miRNAs were grouped, and network analysis was conducted using the software tool String v.10 (https://string-db.org/ accessed on 07/08/2021) with standard settings to visualize networks of target protein–protein interaction.

### 4.5. Statistics

Statistical analyses were performed using RStudio 1.2.1335 (https://cran.r-project.org/ accesses on 06/17/2021), GraphPad Prism 5 software (GraphPad software, La Jolla, CA, USA) and Excel (Microsoft Office, München, Germany). The data are presented as median (10–90 percentiles) and differences between multiple groups were tested using one-way ANOVA with Bonferroni corrected post-hoc analysis or corresponding non-parametric test. For correlation analysis, we used Spearman or non-parametric Pearson testing. Statistical significance was defined as *p* < 0.05.

## Figures and Tables

**Figure 1 ijms-22-10351-f001:**
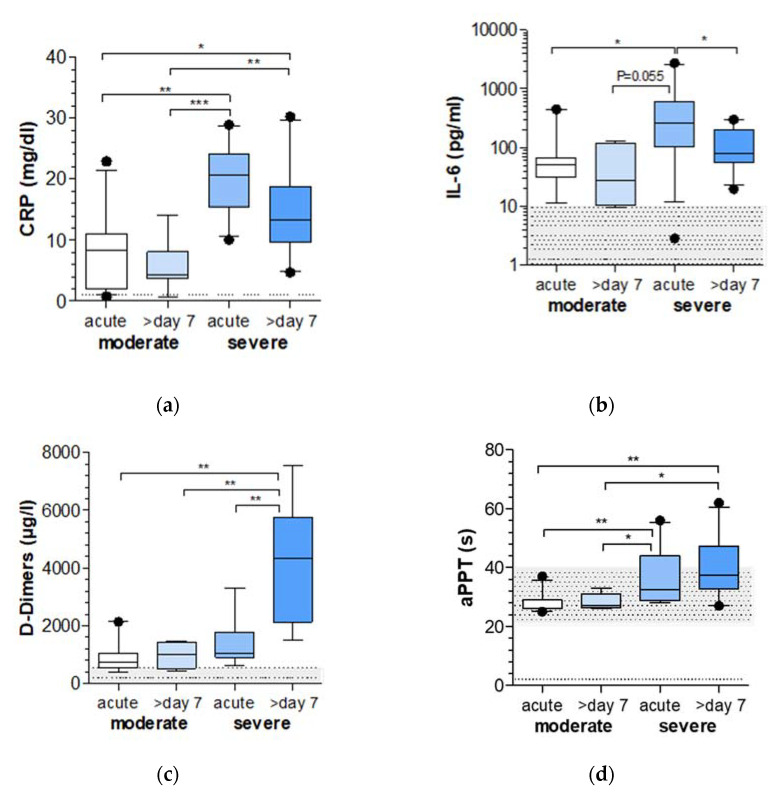
Inflammation and coagulation in COVID-19 patients with moderate and severe respiratory failure. Concentrations of inflammatory biomarkers such as (**a**) C-reactive protein (CRP) and (**b**) interleukin (IL)-6 and the biochemical biomarker (**c**) D-dimer and (**d**) activated partial thromboplastin time (aPTT) were analyzed in the peripheral blood of patients with moderate and severe respiratory failure induced by COVID-19 infection. Biomarkers were assessed during their stay at the infection ward or intensive care unit in the acute stage (day 0–3) and in the course of disease (>7 days). The gray areas indicate the standard ranges. * *p* < 0.05, ** *p* < 0.01, *** *p* < 0.001.

**Figure 2 ijms-22-10351-f002:**
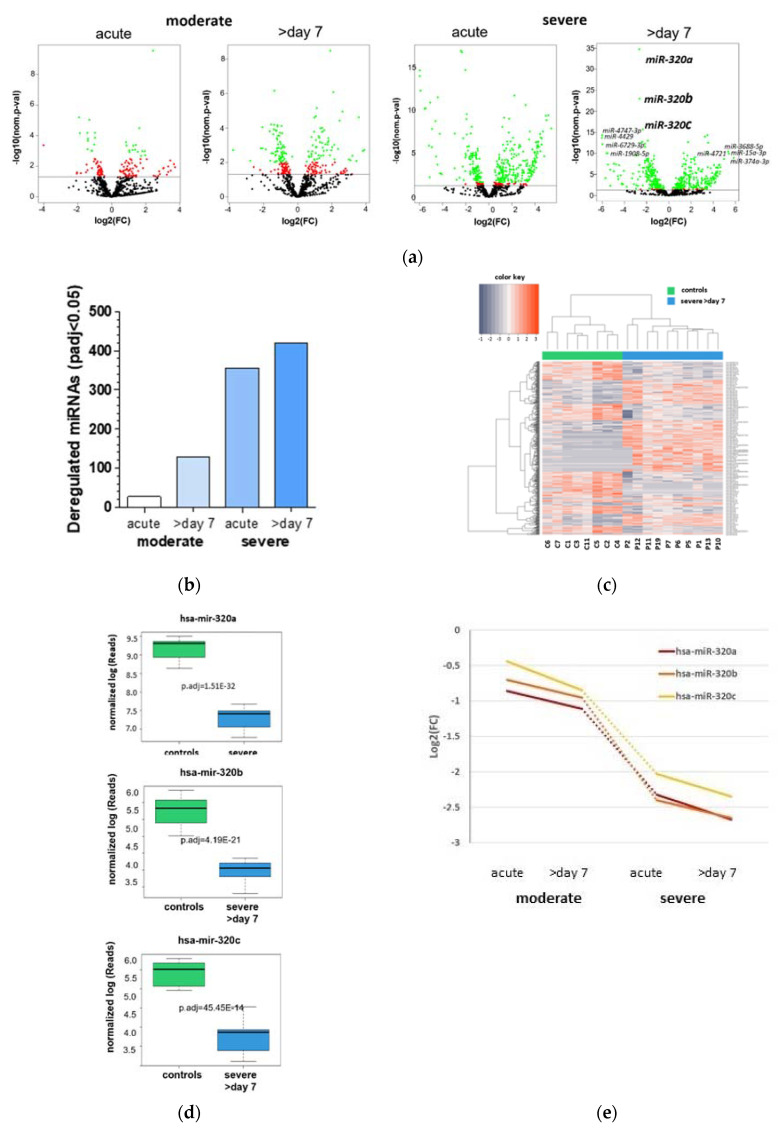
MicroRNA (MiRNA) expression in COVID-19 patients with moderate and severe respiratory failure. (**a**) Volcano plots and (**b**) graph showing number of dysregulated miRNAs (*p*.adj < 0.05) in the peripheral blood of patients with moderate (*n* = 11) and severe (*n* = 10) respiratory failure induced by COVID-19 infection at the acute stage and 7 days later. (**c**) Color blot showing different miRNA expression and (**d**) graphs showing differences in miR-320a, miR-320b and miR-320c expression between patients with severe respiratory failure induced by COVID-19 infection and healthy controls (*n* = 8). (**e**) Fold change expression of miR-320a, miR-320b and miR-320c of patients with moderate and severe respiratory failure induced by COVID-19 infections at acute stage and 7 days later.

**Figure 3 ijms-22-10351-f003:**
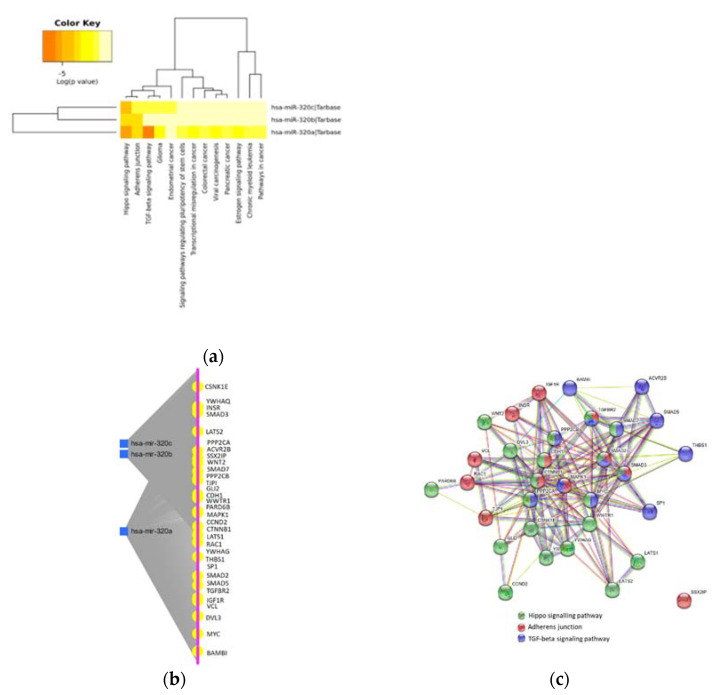
Pathway and target analysis of the miR-320 family. (**a**) Heat map showing Kyoto encyclopedia of genes and genomes (KEGG) pathway analysis of miR-320a, miR-320b and miR-320c using DIANA Tools mirPath v.3 (TarBase v7.0). (**b**) Dissected microRNA (miRNA)-target interactions and functional associations of the three miRs-320 through network-based visual analysis (miRTarBase v8.0) for the identified miRNAs. (**c**) STRING diagram of target protein–protein interaction of the target proteins in Hippo signaling (green bullets), in the transforming growth factor (TGF)-β signaling pathway (blue bullets) and target proteins in the regulation of adherends junction (red bullets). Lines indicate interaction evidence.

**Figure 4 ijms-22-10351-f004:**
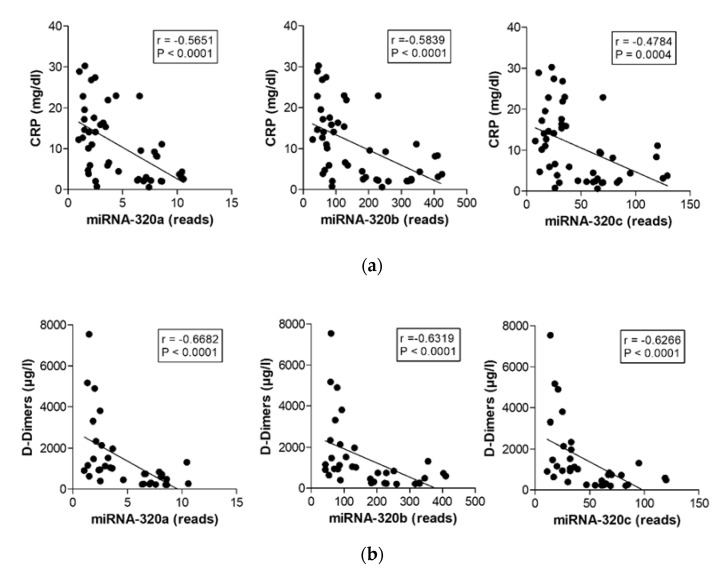
Correlations between expression of the miR-320 family members with markers of inflammation and coagulation. Correlations between miR-320a, miR-320b, and miR-320c and (**a**) CRP, (**b**) D-dimer and (**c**) IL-6 concentration analyzed in the peripheral blood of patients with moderate and severe respiratory failure induced by COVID-19 infection and healthy controls.

**Table 1 ijms-22-10351-t001:** Patient Characteristics.

	Control	COVID-19 with Respiratory Failure	
		Moderate	Severe
*n* =	8	11	10
Age (years)	68 (49–89)	76 (48–91)	69 (52–79)
Length of stay in UKF ^1^ (days)	---	10 (8–43)	26 (14–47)
**Comorbidities** (number of cases)			
Diabetes mellitus	---	4	4
Hypertension	---	9	4
Coronary disease	---	7	6
Chronic kidney disease	---	2	1
Chronic lung disease	---	4	3
Immune suppression	---	2	2
Cancer	---	3	1
Obesity BMI ≥ 30 (kg/m^2^)	---	2	6
**Treatment** (number of cases)			
Anticoagulation	---	11	10
Antibiotic therapy	---	3	10
Catecholamines	---	1	10
Dexamethasone	---	5	1
Remdesivir	---	2	1
Intubation	---	0	10
Non-invasive ventilation	---	3	9
Oxygen without ventilation	---	11	9
Extracorporeal membrane oxygenation	---	0	2
Mortality	---	0	4

^1^ University Hospital Frankfurt.

## Data Availability

The data presented in this study are available on request from the corresponding author. The data are not publicly available due to privacy and ethical concerns.

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
