# Peer review of "The MiR-320 Family Is Strongly Downregulated in Patients with COVID-19 Induced Severe Respiratory Failure"

_ijms, 2021, doi:10.3390/ijms221910351_

Round 1

Reviewer 1 Report

The authors show that MiR-320 family is strongly downregulated in patients with COVID-19 induced severe respiratory failure. This is a nice study with potential importantance in novel biomarkers establishment for COVID-19 patients.

The study would benefit from expansion of discussion on potential relation between miR-320 and inflammatory signaling, particularly IL-6 pathway.

Author Response

Dear Reviewer #1,

Thank you for your kind and very useful comments. We have considered the comment carefully and revised the manuscript in accordance with your suggestion.

Comments and Suggestions for Authors

The authors show that MiR-320 family is strongly downregulated in patients with COVID-19 induced severe respiratory failure. This is a nice study with potential importance in novel biomarkers establishment for COVID-19 patients.

The study would benefit from expansion of discussion on potential relation between miR-320 and inflammatory signaling, particularly IL-6 pathway.

Author response: We thank the reviewer for this valuable comment and added a paragraph about miR-320 and inflammatory signaling including IL-6 to the discussion section. In addition, we have included the correlations of miR-320 family members and IL-6 as figure (Fig. 4c) as well as a short paragraph in the results section.

Reviewer 2 Report

In this manuscript, Duecker and co-workers investigated the differential expression of miRNAs in patients suffering from moderate or severe respiratory failure caused by SARS-COV2 infection. The authors sequenced blood samples derived from control subgroup or patient subgroup and found that several miRNAs were dysregulated in the patient population. Further, the authors focused on miR-320 family which is downregulated in severe patient population compared to control population. The authors found that downregulation of miR-320 family of miRNAs could lead to an increase in inflammation in patients.

The study is correlative in nature, and the authors should tone down the link between miR-320 and inflammation since the observation have not been validated and no mechanistic explanation is provided. 

The authors should also highlight other miRNAs which are more strongly downregulated or upregulated in patient populations (log2 fold change compared to control) and how those miRNAs could contribute to progression of the disease.

Author Response

Dear Reviewer #2,

We thank you for the critical review of our manuscript and your proficient and beneficial comments. We have carefully revised our manuscript in accordance with your suggestions and comments.

In this manuscript, Duecker and co-workers investigated the differential expression of miRNAs in patients suffering from moderate or severe respiratory failure caused by SARS-COV2 infection. The authors sequenced blood samples derived from control subgroup or patient subgroup and found that several miRNAs were dysregulated in the patient population. Further, the authors focused on miR-320 family which is downregulated in severe patient population compared to control population. The authors found that downregulation of miR-320 family of miRNAs could lead to an increase in inflammation in patients.

The study is correlative in nature, and the authors should tone down the link between miR-320 and inflammation since the observation have not been validated and no mechanistic explanation is provided.

Author response: We agree with the reviewer and added a short paragraph about the correlative character of our findings to the discussion section: “However, it is important to keep in mind that the results of the study are correlative in nature and target prediction was performed by computational analysis. Thus, further studies and experiments should be performed to validate our findings.”

The authors should also highlight other miRNAs which are more strongly downregulated or upregulated in patient populations (log2 fold change compared to control) and how those miRNAs could contribute to progression of the disease.

Thank you for your very valuable and useful comment. We highlighted the most down, - and upregulated miRNAs in the manuscript and discussed other miRNAs that were significantly deregulated in our analysis and have already attracted attention in other COVID-19 studies. This further information was added to the result section, the discussion section and we labeled these miRNAs in the volcano plot in Fig. 2a. We additionally added a Supplementary Table (Table S2) with all significantly down, - and upregulated miRNAs in patients with severe respiratory failure.